# Overcoming Fish Defences: The Virulence Factors of *Yersinia ruckeri*

**DOI:** 10.3390/genes10090700

**Published:** 2019-09-11

**Authors:** Agnieszka Wrobel, Jack C. Leo, Dirk Linke

**Affiliations:** 1Department of Biosciences, University of Oslo, 0316 Oslo, Norway; agnieszka.wrobel@ibv.uio.no (A.W.); j.c.leo@ibv.uio.no (J.C.L.); 2Department of Biosciences, School of Science and Technology, Nottingham Trent University, Nottingham NG1 4FQ, UK

**Keywords:** aquaculture, plasmid, fish disease, *Yersinia ruckeri*, virulence factor

## Abstract

*Yersinia ruckeri* is the causative agent of enteric redmouth disease, a bacterial infection of marine and freshwater fish. The disease mainly affects salmonids, and outbreaks have significant economic impact on fish farms all over the world. Vaccination routines are in place against the major serotypes of *Y. ruckeri* but are not effective in all cases. Despite the economic importance of enteric redmouth disease, a detailed molecular understanding of the disease is lacking. A considerable number of mostly omics-based studies have been performed in recent years to identify genes related to *Y. ruckeri* virulence. This review summarizes the knowledge on *Y. ruckeri* virulence factors. Understanding the molecular pathogenicity of *Y. ruckeri* will aid in developing more efficient vaccines and antimicrobial compounds directed against enteric redmouth disease.

## 1. Introduction

Fish are an important protein source for humans, but the production of fish from capture fisheries is not able to meet the increasing demand for protein-rich food due to stresses on ecosystems (overfishing) as well as the growing human population. Aquaculture, the farming of fish, is considered to be more sustainable, but the breeding and raising of large numbers of fish in relatively restricted artificial habitats comes with new problems, including water pollution and diseases.

Infectious diseases caused by bacterial, viral, parasitic and fungal virulent agents are major challenges in aquaculture and currently cumulate to economic losses of about 6 billion USD per year [1], with new diseases emerging constantly [2].

Bacterial diseases, in contrast to viral diseases, play a minor role in recent fish stock losses, thanks to the success of vaccination programs. In recent years, however, the number of outbreaks caused by the devastating pathogen *Yersinia ruckeri*, the causative agent of enteric redmouth disease (ERM), has substantially increased [3]. In contrast to older reports, where the main problems with the disease were caused in the freshwater facilities where fish are hatched and raised in the first weeks, there has more recently been a rapid increase in yersiniosis cases in the seawater life stages of Atlantic salmon [3]. Most of the ERM outbreaks are caused by *Y. ruckeri* serotype O1 biotype 1; however, the incidence of *Y. ruckeri* serotype O1 biotype 2 outbreaks is increasing worldwide [4,5,6,7,8]. Recent emergence of new *Y. ruckeri* serotypes (Table 1) has led to reduced efficiency of the existing licensed vaccines, and to an increased interest in research related to the pathogenesis and virulence of the bacterium. This review will summarize *Y. ruckeri* research in the context of *Y. ruckeri* pathogenesis.

The *Y. ruckeri* serotyping scheme is very complex and it has been revised many times by different authors. It is generally based on serological reactions where *Y. ruckeri* antigenic molecules (lipopolyssachrides, outer membrane proteins, O-antigen) react with antiserum raised against *Y. ruckeri*. The most commonly used serotyping schemes are those introduced by Romalde [12] and Davies [16] in the early 1990s. Davies proposed a scheme based on heat-stable O-antigens where he distinguished five different serotypes (from O1 to O7). Romalde proposed a scheme where differences in lipopolysaccharide and outer membrane protein profiles were taken into account. Four O-serotypes, from O1 to O4, were distinguished. In recent years, new serotypes are emerging (serotype O8 identified in 2016), which has forced researchers to look for new solutions for *Y. ruckeri* treatment.

## 2. Description of *Y. ruckeri*, a Member of the *Yersiniaceae* Family

*Y. ruckeri* is described as a Gram-negative, rod-shaped bacterium of the *Yersiniaceae* family [16],[17]. The cells of this facultative anaerobe can survive in anaerobic and aerobic environments and are generally uniform in morphology. They are rounded, 0.75 µm in diameter and 1–3 µm in length, although differences in cell sizes and cell morphology have been described [9,17,18]. *Y. ruckeri* does not form spores or a capsule. Some *Y. ruckeri* strains are flagellated and consequently exhibit variable motility with peritrichously arranged flagella [19]. Two different biotypes of the bacterium have been described based on the absence or the presence of flagella and their ability to hydrolyze Tween 20/80. Biotype 1 is motile and lipase-positive while biotype 2 is non-motile and lipase-negative. Interestingly, the loss of both motility and lipase activity in biotype 2 is caused by mutations in the flagellar secretion apparatus. Biotype 1 strains isolated in France are longer in size compared to biotype 2 isolates [20]. *Y. ruckeri* can be recovered from the internal organs of infected fish, and can be cultured on various bacteriological media such as Tryptic Soy Agar [21], Nutrient Agar [22], Brain Heart Infusion Agar [7], Columbia Blood Agar [23] and McConkey Agar [24,25]. After 24-48 h of incubation, the bacterium forms smooth, circular, shiny colonies [26]. The cells grow fairly rapidly in a wide temperature range from 22 °C to 37 °C, but best between 22 °C and 25 °C. *Y. ruckeri* can be identified using serological and molecular diagnostic tests [20,23,24,25,26,27,28,29,30,31,32,33,34,35,36]. Recently, a multi-locus variable-number tandem-repeat analysis has been developed as a fast and efficient method for high-resolution genotyping of *Y. ruckeri* isolates [37]. *Y. ruckeri* strains are biochemically uniform regardless of geographical location. *Y. ruckeri* ferments fructose, glucose, glycerol, maltose, mannitol and trehalose, contrary to inositol, lactose, rhamnose, raffinose, sorbitol, sucrose, meliobiose, arabinose and salicin. Most *Y. ruckeri* isolates are Voges-Proskauer-negative and methyl red and citrate-positive. *Y. ruckeri* strains are positive for β-galactosidase, lysine decarboxylase, ornithine decarboxylase and catalase and negative for indole and hydrogen sulfide production, oxidase, phosphatase, urease and phenylalanine deaminase activity [38].

## 3. *Y. ruckeri* as the Causative Agent of Enteric Redmouth Disease

ERM is an acute or chronic bacterial infection in marine and freshwater fish. The mortality rates of ERM are usually low in the initial phase of the disease and then increase rapidly, resulting in severe fish losses. This is especially true when the fish are exposed to stress, for example caused by poor culture conditions [39]. ERM was first documented in the 1950s, when *Y. ruckeri* was isolated from kidney tissue of rainbow trout in the Hagerman Valley in Idaho, USA [17,40]. In 1975, the Fish Health Section of the American Fisheries Society introduced the universal name ‘enteric redmouth disease’, which, together with yersinosis, continues to be the common name used in the literature. The terminology used to characterize ERM pertains to the clinical signs of the disease: the early clinical signs of ERM usually resemble bacterial septicemia caused by other Gram-negative pathogens such as *Aeromonas salmonicida* and *Pseudomonas fluorescens* [41]. The affected fish are darker in color, lose their appetite, become lethargic and swim isolated from the others in regions with slow water flow [19]. Other external signs of the disease include reddening of the mouth (which gave the disease its name), the oral cavity, throat, anus, the base of the fins and the area around the lateral line. This is caused by subcutaneous hemorrhages. Exophthalmia, known as ‘pop-eye’, is another visible sign of the disease. This usually starts unilaterally, but at later stages both eyes can pop out of their sockets. Eventually, exophthalmia causes the eyes to rupture, which results in fish blindness [42]. Internal signs of the infection include petechial hemorrhages in organs such as the liver, pancreas, swim-bladder, stomach and muscles. Hemorrhages and inflammation also occur in the intestine, which is filled with a yellowish, opaque fluid. The kidney and the spleen are often swollen, and the spleen can be black in color [41]. Histological examination of infected rainbow trout tissues typically shows septicemia symptoms in well-vascularized organs such as the kidney, spleen, heart, liver and gills [17,43].

## 4. *Y. ruckeri* Has a Broad Host Range and a Broad Geographic Distribution

*Y. ruckeri* has been isolated from wild and farmed fish as well as from diverse non-fish species [39,44]. Selected non-fish and fish species affected by *Y. ruckeri* are shown in Table 2 and Table 3. Freshwater fish species, particularly salmonids, are most susceptible to *Y. ruckeri* infections. The mortality rate can reach 25–75% when the disease is untreated [26]. ERM has a broad geographic distribution. Following the first documented incidences of *Y. ruckeri* infection in the Hagerman Valley in Idaho, the pathogen was identified in the western states of the USA. From there, the disease has widely spread to other parts of the world, possibly through transport of live farmed fish as shown by the recent multilocus variable-number tandem repeat analysis [26,45]. In 1971, McDaniel documented the presence of the disease in an area of the Rocky Mountains in Wyoming [46]. During the same year, the disease occurred in other states including Alaska, Oregon, Utah and Washington [46]. In 1978, Bullock et al. reported the presence of ERM in 18 states [9]. In 1972, the disease was reported in caged rainbow trout in a southern Saskatchewan lake in Canada [47]. In the following years, the disease spread to Australia [48] and various European countries such as Denmark [49], France [50], Germany [41], Italy [51], Norway [52], Poland [53] and Croatia [54]. ERM was also encountered in Romania [55], New Zealand [56], South Africa [57] and Chile [58]. Today, the disease is spread over the entire globe with the most recent outbreaks in Australia, Norway and Scotland [15].

## 5. *Y. ruckeri* Pathogenicity Factors

It is well known that *Y. ruckeri* is responsible for economic losses in rainbow trout and Atlantic salmon farming production. Nevertheless, there are still a limited number of studies examining the virulence mechanisms of this pathogen. This may be the result of the early success of the ERM vaccine in the 1960s, the complex taxonomy of the pathogen, as well as scarce information on *Y. ruckeri* serotypes [59,83]. The virulence of *Y. ruckeri* is influenced by several factors, such as iron availability, temperature, pH and osmolarity [83]. Furthermore, it is generally believed that the virulence mechanisms and determinants of *Y. ruckeri* are similar to those in encountered in other members of the *Yersiniaceae* family [84]. Indeed, many researchers re-used models and techniques that were already applied to study other members of the *Yersiniae*. Various approaches have been used to identify and then further study the virulence-related genes of *Y. ruckeri* [83].

Molecular approaches for identifying virulence factors include the in vivo expression technology (IVET) approach and signature-tagged mutagenesis (STM) [85,86]. The IVET is based on the identification of genes that are expressed in vivo in the host during the infection process. In contrast, STM is issued to screen for *Y. ruckeri* mutants that survive in vitro, but not necessarily in vivo, meaning that the inactivated gene is essential for survival in the host but not necessarily directly for infection. Other approaches are based on the bioinformatics prediction of the role of virulence factors, and they include proteomic, transcriptomic and genomic analyses [83,87,88]. Although all of these methods have limitations, they have helped to discover many *Y. ruckeri* virulence factors. A summary of the key factors identified as of today is depicted in Figure 1.

### 5.1. Y. ruckeri Toxins

#### 5.1.1. *Y. ruckeri* Protease 1 (Yrp1)

Yrp1, *Y. ruckeri* protease 1, is a 47-kDa serralysin metalloprotease that was identified by Secades and Guijarro in 1999 [89]. Yrp1 is secreted by an ATP-dependent type 1 secretion system (T1SS), similarly to hemolysin A from uropathogenic *Escherichia coli* [90]. The secretion system comprises three genes, *yrpDEF*, and an additional protease inhibitor *inh* [91]. Yrp1 is produced by some of the most virulent *Y. ruckeri* strains, named Azo^+^ strains (where *azo* is the gene for Yrp1), at the end of the exponential growth phase [89]. Expression of Yrp1 is controlled by osmolarity and temperature: highest expression occurs at 18 °C and under low osmotic pressure [92]. Yrp1 protease has been implicated in virulence [91]. It digests a wide range of matrix and muscle proteins, such as laminin, fibronectin, actin and myosin, causing the typical clinical symptoms of ERM. Finally, inactivated Yrp1 is protective against ERM when delivered as an immunogen [83,92].

#### 5.1.2. *Y. ruckeri* Peptidases (YrpAB)

*yrpAB* are two adjacent genes encoding two peptidases from the U32 family. Both genes, identified by IVET, share a high sequence identity and the same genetic arrangement as peptidases present in other members of the *Enterobacterales* group [86,93]. The *yrpAB* operon expression is dependent on the presence of peptone in the culture media as well as the presence of oxygen. The expression of the *yrpAB* operon is highly upregulated under microaerobic conditions, which are encountered in fish gut, indicating that both peptidases may be expressed in the gut tissue. Interestingly, an LD_50_ experiment showed that at least one of the peptidases, YrpA, is involved in *Y. ruckeri* virulence. A deletion mutant of *yrpA* showed attenuated virulence (infective dose 7.1 × 10^4^ colony forming units) compared to the wild-type (WT) strain (1.73 × 10^2^ colony forming units) [93].

#### 5.1.3. *Y. ruckeri* Pore-Forming Toxin, *yhlBA*

The *yhlBA* cluster consists of two adjacent genes termed *yhlA* and *yhlB* identified by IVET [86]. *yhlA* encodes a hemolysin while *yhlB* is involved in *yhlA* activation and secretion [94]. Together, they constitute a type Vb secretion system [95]. Both genes have a high sequence similarity to genes encoding type Vb secretion systems with hemolysin activity in *Serratia* sp. Expression of the operon is regulated by temperature and iron availability and leads to cytolysis and hemolysis, e.g., of erythrocytes and BF-2 fish cells [94]. A high level of *yhlBA* expression was observed under limited iron availability and at 18 °C. The presence of the *yhlBA* operon in *Y. ruckeri* strains from different geographic areas and sources indicates an important role for this cluster in *Y. ruckeri* virulence [83],[94]. Interestingly, a hemolysin gene cluster is also present in human-pathogenic *Yersiniae*, but its role in virulence has not been confirmed [94,96].

#### 5.1.4. *Y. ruckeri* Phospholipase

Phospholipases have the potential to act as exotoxins that disrupt host cell membranes [97], and the phospholipase activity of *Y. ruckeri* biotype 1 has been implicated in virulence functions. Phospholipase activity is found in the extracellular fraction of *Y. ruckeri* cultures, and the heat-labile products in the fraction are toxic to fish and show dermatotoxic effects in a rabbit skin model [98],[99]. The secretion of the phospholipase is dependent on the flagellar secretion machinery [100,101,102]. However, as the emerging biotype 2 does not have phospholipase activity, this cannot be considered an essential virulence factor.

#### 5.1.5. Antifeeding Prophage 18 (Afp18)

Afp is a prophage-encoded toxin delivery system that shows similarities to R-pyocins of *Pseudomonas aeuroginosa,* and the *Photohabdus luminescens* virulence cassette [103,104]. Afp is present in Gram-negative and Gram-positive bacteria as well as in archaea. It resembles the tail of T4 bacteriophage that is composed of a central tube covered by a contractile sheath and a baseplate with fibers, which together probably constitute a type VI secretion system [105]. Afp encoded by the pADAP plasmid of *Serratia entomophila* was shown to play a role in delivery of a toxin, Afp18 [106]. In *Y. ruckeri,* Afp18 contains a glycosyltransferase domain at the C-terminus. After injection of the recombinant glycosyltransferase domain into zebrafish embryos, the development of zebrafish stopped, indicating a direct role in virulence for Afp18 [104].

### 5.2. Y. ruckeri Secretion Systems

#### 5.2.1. *Y. ruckeri* Type III Secretion System (T3SS)

T3SSs are used by many Gram-negative pathogens to deliver toxins into host cells. The T3SS, also known as the injectisome, consists of a basal body that spans both inner and outer membrane, and a needle that protrudes out of the cell and makes direct contact between the bacterial cell and a host cell. The T3SS encodes three categories of proteins: structural proteins, chaperones and effector proteins [107].

A complete T3SS, the *ysa* locus, was recently found in the genome of *Y. ruckeri* SC09 [108]. *Y. ruckeri* SC09, isolated from a moribut (*Ictalurus punctatus)*, possesses the largest genome within the species, with a length of 3.9 Mb encoding 3651 predicted protein-coding sequences. The *ysa* locus is found on the chromosome and shows similarities in gene content, gene sequence and gene arrangement with the *Salmonella enterica* pathogenicity island 1 and the chromosome-encoded T3SS from *Y. enterocolitica* biotype 1B. It is thus likely that the T3SS of SC09 is required for intracellular survival in fish macrophages, though this has not been experimentally verified [108].

#### 5.2.2. *Y. ruckeri* Type IV Secretion Systems (T4SSs)

T4SSs are found in Gram-negative and Gram-positive bacteria, and in some archaea [109]. T4SSs are used by bacteria to transport macromolecules, such as proteins, protein complexes, protein-DNA complexes or DNA molecules across the bacterial cell membrane. Three major groups of T4SSs have been identified based on their function [110]. The first group, called conjugative T4SSs, transfers DNA from one cell to another by direct contact, in a process called conjugation. These T4SSs can be found on self-transmissible plasmids. They can also be a part of a chromosome as part of conjugative transposons. The second group of T4SSs is responsible for DNA uptake and release into the extracellular milieu. Two examples are the ComB system found in *Helicobacter pylori* (DNA uptake) and Gonococcal Genetic Island found in *Neisseria gonorrhoeae* (DNA release). The third group of T4SSs transfers virulence proteins and protein complexes into host cells, and thus plays an important role in virulence. This group is exemplified by T4SSs from important human pathogens, such as *H. pylori, Bordetella pertussis* and *Legionella pneumophila* [110].

T4SSs have been identified in different *Y. ruckeri* strains, and interestingly are absent in human-pathogenic *Yersiniae*. The *tra* operon of *Y. ruckeri* 150 was identified by IVET [86]. For *Y. ruckeri* SC09, it was suggested that the T4SS might be required for pathogen survival in fish macrophages [108]. The *tra* operon was shown to resemble the *tra* operon of the pADAP plasmid from *S. entomophila*, both in sequence and gene organization, and some genes of the *tra* operon display sequence similarity to the Dot/Icm proteins of *L. pneumophila*. A mutant of the *traI* gene of *Y. ruckeri* 150 was reported to be less virulent than the WT strain [111]. The expression of the *tra* operon is regulated by temperature and the availability of nutrients. The highest expression of the T4SS genes was reported at 18 °C, with limited nutrients. In contrast to previous reports, where the T4SS genes were reported to be localized on the chromosome, long-read sequencing technologies showed that it is actually encoded on a large plasmid, pYR4, of the *Y. ruckeri* NVH_3758 strain [112]. Based on this study, the previously reported chromosomal localization of the operon, based on short-read sequencing, should be revised. A recent comparative analysis of genome sequences of different *Y. ruckeri* O1 serotype strains (150, CSF007-82, ATCC29473) showed 268 O1-specific genes present in all three genomes, and genes of the T4SS were among these shared genes [113].

#### 5.2.3. *Y. ruckeri* Type V Secretion Systems (T5SSs)

T5SSs or autotransporters are the most widespread secretion systems in Gram-negative bacteria. T5SSs contain a translocator unit, consisting of an outer membrane β-barrel domain and possibly some accessory domains, and an extracellular effector region or passenger [114]. Currently, five subtypes of T5SSs are recognized [95]. The only T5SS subtype present in *Y. ruckeri* is the type Ve or inverse autotransporter (IAT) subtype, which comprises virulence-associated adhesins that are also present in human-pathogenic *Yersiniae* [115]. IATs are found in *Proteobacteria, Chlamydia, Cyanobacteria* and *Planctomycetes* [116,117]. To date, they have been classified into four different sub-families: intimin, invasin, FdeC-type adhesins and two-partner IATs [117]. The two best-studied examples of the IATs are intimin of *E. coli* and invasin of *Y. enterocolitica* and *Y. pseudotuberculosis*, which play an important role in bacterial attachment to host tissues.

Two IAT genes are present in the genome of *Y. ruckeri* strains, termed *yrInv* (for *Y. ruckeri* invasin) and *yrIlm* (for *Y. ruckeri* invasin-like molecule) [118]. These IATs are variably found in *Y. ruckeri* strains of serotype O1 and serotype O2, isolated from rainbow trout and Atlantic salmon. The presence of the IATs in *Y. ruckeri* strains from different geographical regions and different sources may indicate a role in virulence, though their effects in vivo are yet to be tested. Both *yrInv* and *yrIlm* display a domain organization similar to well-described IATs: an N-terminal signal peptide followed by a periplasmically located LysM minidomain, which mediates oligomer formation and binding to peptidoglycan [119]. The β-barrel domain is between the periplasmic domain and the exported passenger, which contains several immunoglobulin (Ig)-like domains and a C-terminal lectin-like domain. YrInv and YrIlm show, respectively, 53% and 40% sequence similarity with InvA of the human pathogen *Y. pseudotuberculosis*. YrIlm is significantly longer in sequence compared to YrInv and comprises ~19 practically identical Ig-like domains, with some variation in repeat numbers between different strains. YrInv, by contrast, only contains 3 Ig-like domains with more divergent sequences. Expression of both genes is regulated by environmental factors, including temperature, salt concentration, iron availability and the presence of oxygen [118].

#### 5.2.4. Type IV Pili

Type IV pili are multi-subunit, retractable surface appendages that can bind to surfaces and confer twitching motility. In addition, type IV pili can also have other functions such as DNA uptake or biofilm formation [120]. Type IV pili are secreted by a large complex spanning both membranes of Gram-negative bacteria and are related to T2SSs. Some *Y. ruckeri* strains contain a type IV pilus gene cluster originally thought to reside on the chromosome, but more recent analysis suggests that this locus (along with a separate T4SS) is located on a plasmid [111,112]. As the *pil* locus encoding the type IV pilus is adjacent to the T4SS *tra* locus, these may be functionally coupled. The role of the *pil* locus in virulence has not been studied, but as it is genetically linked to the *tra* operon, which does play a role in pathogenicity, a virulence function for *pil* can also be inferred.

### 5.3. Operons Involved in Ion and Amino Acids Transport

#### 5.3.1. Iron Acquisition System

Iron acquisition systems aid many bacterial pathogens in colonizing and invading host tissues effectively. *Y. ruckeri* has such a system and produces the iron-chelating enterobactin-like siderophore ruckerbactin [19]. Siderophores are low-molecular-mass and high-affinity iron-chelating molecules [19]. They are categorized into four major groups: catecholates (enterobactin), hydroxamates (ferrioxamine), carboxylates (rhizobactin) and a mix of different types (pyoverdine) [121]. During bacterial infection, when iron availability is limiting for bacterial growth, the siderophores produced by bacterial cells are excreted into the extracellular environment, where they form a siderophore-Fe^3+^complex, which is then transported back to the bacterial cell and into the cytoplasm through a specific bacterial outer membrane receptor. Once inside the bacterial cell, the Fe^3+^ is reduced to Fe^2+^ in order to release free iron, which can then be utilized in metabolic reactions [19].

The genes involved in production of ruckerbactin are upregulated during the infection process and are highly expressed at 18 °C, but not at 28 °C, which is the optimal temperature for the pathogen’s growth [86]. Most of the ruckerbactin gene cluster has high sequence similarity with the *E. coli* enterobactin cluster, and their gene organization is similar. There is also a high degree of similarity between the ruckerbactin receptor and the ferrichrysobactin receptor of *Dickeya dadantii* (formerly known as *Erwinia chrysanthemi*), but not with the enterobactin receptor of *E. coli* [86].

#### 5.3.2. The *cdsAB* Operon

The *cdsAB* operon was identified using IVET [86]. The operon is involved in the uptake and degradation of L-cysteine. The operon consists of two genes, *cdsA* and *cdsB*, that encode a cysteine permease and an L-cysteine desulfidase, respectively. The *cdsAB* cluster is found in several groups of Gram-negative bacteria, but not in *Y. pseudotuberculosis* or *Y. pestis* [122]. However, it is present in *Y. ruckeri* strains from different geographical locations and origins. In vivo assays and LD_50_ experiments showed that the *cdsAB* cluster is required to achieve full virulence. The underlying mechanism for the virulence effects is not known; it has been speculated that the genes are involved in the assembly of virulence-related proteins that contain iron-sulfur clusters, or in the release of H_2_S to induce host signaling [122].

#### 5.3.3. *znuABC* and BarA-UvrY

The *znuABC* operon is composed of three genes, encoding a zinc-binding transporter, a membrane permease, and an ATPase. The genetic organization of the operon is similar to its counterparts found in other Gram-negative bacteria, such as *E. coli* and *Salmonella* Typhimurium [123,124]. The *znuABC* operon is likely to be involved in the transport of zinc, a molecule required for the pathogen’s survival. This is supported by the fact that the growth of *E. coli znuABC* mutants was restored in a metal-deficient medium when the *znuABC* operon of *Y. ruckeri* was supplied on a plasmid. Concomitantly, a *znuABC* mutant of *Y. ruckeri* had low virulence and survived poorly in rainbow trout [125]

In uropathogenic *E. coli*, the BarA-UvrY two-component system has been suggested to regulate switching between glycolysis and gluconeogenesis [126]. The UvrY component of the BarA-UvrY system plays the role of a response regulator, and UvrY was demonstrated to contribute to *Y. ruckeri* pathogenicity. Mutations in *uvrY* led to reduced invasion of EPC epithelial fish cells and increased sensitivity to oxidative stress, which in turn impaired the survival of *Y. ruckeri* in rainbow trout [127].

### 5.4. Other Virulence Factors

#### 5.4.1. Genes Upregulated at 18 °C

A recent study identified putative virulence genes upregulated at 18 °C, a temperature at which most *Y. ruckeri* outbreaks occur. A *Tn*5–based transposon cassette containing the promoterless *lux-lac* operon was randomly inserted into the genome of *Y. ruckeri*, causing interruption of genes and using their cognate promotors. As a result of that, 168 clones were identified that had higher β-galactosidase activity at 18 °C than at 28 °C. The interrupted genes included, among others, genes there were involved in the synthesis of legionaminic acid (a component of the lipopolysaccharide (LPS) structure), the *yrp1* metalloprotease described above, genes regulated by environmental changes (for example, the diguanylate cyclase, a glycosyltransferase and a S-adenosyl methionedependent methyltransferase), and genes induced under osmotic shock (for example *osmY*). All these genes were previously implicated in virulence in other organisms [128].

#### 5.4.2. Heat-Sensitive Factor (HSF)

HSF was identified in *Y. ruckeri* in 1990 by Furones et al. [129]. A comparative study of cell extracts between virulent and avirulent *Y. ruckeri* strains exhibited a correlation between the production of HSF by virulent strains. Cell extracts of *Y. ruckeri* strains of serotype I, when injected into fish, caused mortalities only in HSF-positive strains. HSF is likely to play a role in virulence. It was proposed that HSF can mask cell surface antigens, and thus, confer resistance to phagocytic killing. However, a recent study showed that HSF is an alkyl sulphatase, encoded by the *yraS* gene. The alkyl sulphatase catalyzes the hydrolysis of SDS, but is not directly linked to *Y. ruckeri* virulence [22].

#### 5.4.3. LPS

LPS is a glycolipid present in the outer leaflet of the outer membranes of Gram-negative bacteria and has been implicated in virulence in numerous species. The *lpxD* gene is one of the nine genes involved in biosynthesis of the lipid A moiety of LPS [130]. In-frame deletion of the *lpxD* gene in *Y. ruckeri* led to a production of an attenuated *Y. ruckeri* strain. The *lpxD* mutant, when administered as a vaccine by either immersion or injection to rainbow trout, elicited good protection responses against WT *Y. ruckeri.* The LPS-deficient mutant was suggested to be a good vaccine candidate against *Y. ruckeri* infections in rainbow trout [83,130].

#### 5.4.4. Flagella

The *flhDC* operon of *Y. ruckeri* is a master regulator of the flagellar secretion apparatus that has been shown to regulate flagella production and phospholipase secretion. Mutation in the *flhDC* operon prevents motility and phospholipase activity, and additionally, absence of *flhD* results in transcriptional changes in a variety of genes. In competition assays, the *flh∆D* mutant of *Y. ruckeri* was reported to be more virulent than the WT strain. Examination of spleen in rainbow trout revealed an increased cell density of the *flh∆D* mutant compared to the WT [101]. A very recent study demonstrated that flagella expression is regulated by the Rcs system, which exerts its effect through modulating the expression of *flhDC* [102]. Flagellar synthesis is repressed within the host, but de-repressed upon host death. The repression of flagella is presumably to avoid detection by the host immune system. Strains with mutations in the *rcsB* gene produced flagella under conditions where flagellar synthesis is normally repressed and were attenuated in a fish infection model [102].

#### 5.4.5. Plasmids

Plasmids play an important role in the virulence of human-pathogenic of *Yersiniae*. The pYV-type plasmid (YV stands for *Yersinia* virulence) is the best-characterized plasmid in the *Yersiniae*. It is shared among the three human-pathogenic *Yersiniae* (*Y. pseudotuberculosis, Y. enterocolitica* and *Y. pestis*). Removal of this plasmid was shown to result in the complete loss of virulence. The pYV plasmid is a ~70-kb virulence plasmid. In *Y. pseudotuberculosis* and *Y. enterocolitica,* it encodes a T3SSs and a T5SS protein named YadA (*Yersinia* adhesin A) [131,132]. The notorious *Y. pestis* typically carries two additional plasmids, pPla (encoding the plasminogen activator) and pFra (carrying the genes for murine toxin and the F1 antigen) [133]. The presence of further plasmids in *Y. pestis* has been reported; however, they are strain-specific and not very well characterized [134,135].

De Grandis and Stevenson examined the presence of plasmids in *Y. ruckeri* strains representing serotypes I and II [136]. Strains of serotype I contained a large and a small plasmid, while strains of serotype II contained only small plasmids, regardless of the source of the isolate. These results agree with those obtained by others [10,137,138]. The apparent number of plasmids in *Y. ruckeri* strains can differ depending on the applied plasmid isolation method [138]. The presence of large plasmids was suggested to be related to temperature, as the strains carrying the large plasmids were unable to grow at 37 °C. *Y. ruckeri* strains that carried the large plasmids exhibited distinct colony morphology and serological reactions, indicating that the large plasmid might encode genes related to these features [137]. Large plasmids have received a lot of attention due to their possible correlation with *Yersiniae* virulence [139]. Guilvout et al. examined the plasmid profile of 18 *Y. ruckeri* strains isolated from various ERM outbreaks. Strains of serotype II and V contained four plasmids each with both high and low molecular weights (72 MDa, 62 MDa, 32 MDa, 25 MDa), while strains of serotype I carried only one plasmid (62 MDa) [140]. Southern hybridization experiments between 62-MDa plasmids of *Y. ruckeri* strains of different origin showed a high level of similarity. However, the 62-MDa plasmid did not show any DNA similarity with the pYV plasmids from other *Yersiniae,* indicating a different origin and possibly, different functions.

To date, the function of *Y. ruckeri* plasmids is not clear. Large plasmids present in *Y. ruckeri* often carry antibiotic resistance genes. De Grandis and Stevenson tested susceptibility of *Y. ruckeri* strains to 23 antimicrobial agents [141]. All tested *Y. ruckeri* strains showed a similar antimicrobial susceptibility pattern except for two strains, which carried a 36-MDa plasmid and a 52-MDa plasmid and exhibited resistance to sulfonamides and tetracycline. Conjugation experiments showed that one of these two plasmids encoded antibiotic resistance genes that could be transferred to *E. coli* and to other *Y. ruckeri* strains.

Welch et al. showed a comparative DNA sequence analysis between the pIP1202 plasmid of *Y. pestis* strain IP275 and multi-drug resistance plasmids of *Y. ruckeri* YR1 (pYR1) and of the *S. enterica* serotype Newport SL254 (pSN254) [142]. The plasmid backbones were highly similar between those pathogens, indicating that the plasmids had a common ancestor. The common region contained genes involved in plasmid replication and type IV conjugative transfer. A recent study suggested that plasmids can be linked to *Y. ruckeri* virulence. A plasmid, pYR4, was found and analyzed, that—similarly to pYR3 from earlier studies—encodes both a type IV pilus and a T4SS (see above) that are implicated in virulence [112].

#### 5.4.6. TcpA

TcpA, recently identified by Liu et al., is a virulence protein encoded by integrative and conjugative elements [143]. TcpA is present in Y. ruckeri SC09, but absent in other Y. ruckeri strains. In vivo and in vitro data showed TcpA involvement in virulence. TcpA disrupts Toll like receptor signaling in host cells and thus facilitates the escape from the host immune response, and therefore, enables bacterial survival.

#### 5.4.7. Outer Membrane Proteins (OMPs)

OMPs are partly surface-exposed membrane proteins found in Gram-negative bacteria. They adopt a transmembrane β-barrel fold with eight to 26 strands, and play an essential role in nutrient acquisition, cell survival and host-pathogen interactions [144]. The OMPs of *Y. ruckeri* are very poorly characterized and there is little knowledge about their involvement in virulence. In recent years, however, OMPs have been discussed as promising vaccine candidates in fish. A recent study explored the protective capabilities of outer membrane porin F (OmpF) in *Y. ruckeri* [145]. OmpF is one of the major *Y. ruckeri* OMPs. Phylogenetic and multiple alignment analysis of OmpF demonstrated high sequence identity of OmpF within the *Yersiniae* (80.8–86.8%), compared to other *Enterobacterales* (58.8–76.4%), indicating that OmpF is a promising vaccine candidate. Indeed, recombinant OmpF injected into channel catfish (*I. punctatus*) enhanced the immune response by increasing, among others, lysozyme activity, complement C3 activity and serum antibody levels [145].

A recent study of *Y. ruckeri* strains isolated from Atlantic salmon and rainbow trout in Scotland demonstrated a high level of diversity between Atlantic salmon and rainbow trout *Y. ruckeri* isolates [15]. Specifically, the OMPs profile of aerobically grown *Y. ruckeri* isolates recovered from Atlantic salmon and rainbow trout was clearly different. The OMP profile of *Y. ruckeri* isolates recovered from Atlantic salmon was diverse: the isolates belonged to three different profile types (2a, 2c and 3a). In contrast, the OMP profiles of *Y. ruckeri* isolates recovered from rainbow trout were highly homogenous: most of the isolates were assigned to OMP type 3a [15].

A global proteomics analysis of *Y. ruckeri* strains of biotype 1 and biotype 2 grown under standard culture conditions was recently performed, identifying a wide number of OMPs (OmpA, OmpC, OmpF and OmpW) in addition to other putative virulence factors [88]. In a similar study, protein identification and quantification under standard and iron-limited conditions of motile and non-motile *Y. ruckeri* strains was performed. Sixty-one proteins were differentially expressed between iron-limiting and normal growth conditions; among the upregulated proteins were multiple OMPs related to iron and copper transport [87]. Similarly, a recently performed comparative bioinformatic and proteomic analysis of the OM proteome of *Y. ruckeri* strains recovered from Atlantic salmon and rainbow trout identified genes related to host specificity. Eighty-four OMPs were identified that were expressed under in vitro standard culture conditions. Sixteen OMPs were exclusively assigned to *Y. ruckeri* isolates from rainbow trout, including, e.g., filamentous hemagglutinin, metallopeptidases, a siderophore receptor and components of the T3SS that were predicted to be involved in adhesion to host cells, iron uptake and intracellular survival, respectively. Five proteins were exclusive for *Y. ruckeri* isolates of Atlantic salmon and these included proteins involved in autotransport, phospholipid-hydrolyzing enzymes and proteins involved in uptake of the ruckerbactin [146].

#### 5.4.8. Biofilm as a Virulence Factor

*Y. ruckeri* can form biofilms in aquatic environments [147,148,149]. *Y. ruckeri* adheres to solid supports such as PVC, fiberglass, concrete, wood and other materials commonly found in fish farms. Surface roughness plays an important role in *Y. ruckeri* adherence to surfaces. Moreover, with longer incubation times, biofilm formation is more pronounced [148]. Finally, it has been suggested that biofilm formation by *Y. ruckeri* is key to recurrent infections in rainbow trout aquaculture, based on the ability of bacteria in biofilms to persist in circulating water systems [147].

The efficiency of biofilm formation varies from strain to strain and seems to be correlated with the presence of flagella-mediated motility. Environmental *Y. ruckeri* isolates exhibit a higher ability to adhere to surfaces than the reference strain, and thus, they are more likely to form a biofilm. Surface-associated *Y. ruckeri* cells are more tolerant to antibiotics than planktonic cells [148]. This can be both a direct effect of biofilm formation, or can in part be explained by differences between the outer membrane pattern of surface-associated *Y. ruckeri* strains and planktonic cells [147,149].

*Y. ruckeri* can adhere to and invade into various fish cell lines [150,151]. Application of inhibitors targeting host mechanisms significantly decreased the susceptibility of fish cell lines to be invaded by *Y. ruckeri* isolates of biotype 1 and biotype 2. Overall, biotype 1 strains were more adhesive and more invasive compared to biotype 2 strains. This finding indicates that *Y. ruckeri* targets diverse host’s mechanisms to gain entry into host cells. Additionally, the presence of flagella seems to be crucial for bacterial adhesion and biofilm formation and for entry into the host. The role of quorum sensing on the regulation of virulence factors, biofilm formation and swimming motility of *Y. ruckeri* was recently studied [152]. The supplementation of *Y. ruckeri* bacterial culture with a signaling molecule (3-oxo-C8-homoserine lactone) significantly increased the swimming motility and biofilm formation of *Y. ruckeri* isolates in vitro, while quorum quenching ((QQ) = interruption of bacterial communication) bacteria had the opposite effect on biofilm formation and motility of *Y. ruckeri* strains. This suggests that QQ bacteria can influence the gene expression of *Y. ruckeri* strains. Interestingly, such QQ bacteria are present on fish gills, in the fish intestine and on fish skin. Because *Y. ruckeri* enters the host via the secondary lamellae of the gills, it is possible that QQ bacteria might interfere with the pathogen by downregulating the expression of *Y. ruckeri* virulence genes, and therefore, impede the entry of the pathogen into the fish host [152,153,154].

## 6. Discussion

The current knowledge on *Y. ruckeri* virulence factors is incomplete and fragmented. In this review, we have tried to summarize the state of the art. Most studies of *Y. ruckeri* virulence are either descriptive or rely on genomics and proteomics methods to identify virulence factors based on sequence comparison with better-studied pathogenic species. Future work will have to experimentally address the molecular mechanism of the identified putative virulence factors to verify these findings, and to correlate these findings with the knowledge on the different known serotypes. A deeper understanding of the virulence mechanisms will be the foundation for the development of new antimicrobial strategies against this economically important fish pathogen, to complement existing vaccines and to avoid excessive use of antibiotics.

## Figures and Tables

**Figure 1 genes-10-00700-f001:**
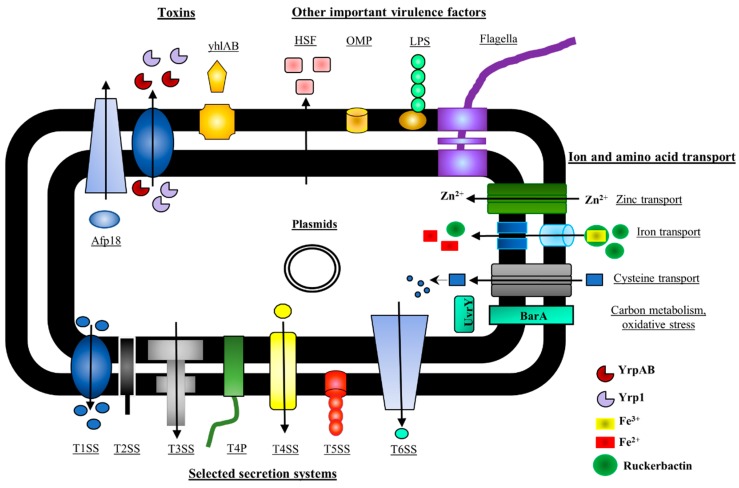
Schematic representation of the selected virulence factors in *Y. ruckeri*, modified from [83]. Secretion systems are depicted in different colors: blue (T1SS; type I secretion system), black (T2SS; type II secretion system), grey (T3SS; type III secretion system), dark green (T4P; type IV pili), yellow (T4SS; type IV secretion system), red (T5SS, type V secretion system) and light blue (T6SS; type VI secretion system). A variety of *Y. ruckeri* toxins are transported via T1SS such as proteases (Yrp1, in light purple) and peptidases (YrpAB, in brown), and via a T5SS a pore forming toxin (yhlBA, in orange). The Antifeeding Prophage 18 (Afp18) secreted via a T6SS is depicted in light blue. Other important virulence factors include OMPs (outer membrane proteins—in dark yellow), LPS (lipopolysaccharides—in green), flagella (in purple) as well as HSF (heat sensitive factor—in light red), and zinc and cysteine transporters. L-cysteine (light blue squares) is taken up by the cell with the help of cysteine permease (*cdsA*), while L-cysteine breakdown is accomplished inside the cell by L-cysteine desulfidase (*cdsB*). Ruckerbactin (in dark green) is involved in the transport of iron across the cell membrane. In the iron limiting conditions, ruckerbactin forms a complex with Fe^3+^ outside a cell. Then, the ruckerbactin- Fe^3+^ complex is transported across the cell membrane to be reduced to Fe^2+^, and finally to release free iron that can be used by the cell.

**Table 1 genes-10-00700-t001:** Serotyping scheme for *Y. ruckeri* between 1977 and 2016 [9,10,11,12,13,14,15].

	Year						
	1977	1978	1984	1988	1990	1993	2016
Serotype name	I	I	I	-	O1	O1a	O1
	II	II	II	II	O2	O2a,b,c	O2
	-	III	III	III	O1	O1b	O1
	-	-	IV	-	-	-	-
	-	-	V	V	O5	O3	O5
	-	-	VI	VI	O6	O4	O6
	-	-	-	-	O7	-	O7
							O8

**Table 2 genes-10-00700-t002:** Selected fish species susceptible to *Y. ruckeri* infections [44,59].

Common Fish Name	Scientific Name	Reference
Arctic char	*Salvelinus alpinus*	[60]
Atlantic cod	*Gadus morhua*	[61]
Atlantic salmon	*Salmo salar*	[15,60,62]
Bighead carp	*Aristichthys nobilis*	[63]
Burbot	*Lota lota*	[64]
Brook trout	*Salvelinus fontinalis*	[21,65]
Brown trout	*Salmo trutta*	[66,67]
Coalfish	*Pollachius virens*	[68]
Coho salmon	*Oncorhynchus kisutch*	[69]
Common carp	*Cyprinus carpio*	[70]
Chinook salmon	*Oncorhynchus tshawytscha*	[65]
Cisco	*Coregonus artedii*	[21]
Cutthroat trout	*Salmo clarkii*	[71]
Eel	*Anguilla anguilla*	[41]
Fathead minnow	*Pimephales promelas*	[68]
Goldfish	*Carassius auratus auratus*	[72]
Muksun	*Coregonus muksun*	[73]
Nile tilapia	*Oreochromis niloticus*	[74]
Perch	*Perca fluviatilis*	[67]
Peled	*Coregonus peled*	[73]
Rainbow trout	*Oncorhynchus mykiss*	[17,38,41,66,67,71,75]
Rudd	*Scardinius erythrophthalmus*	[76]
Sockeye salmon	*Oncorhynchus nerka*	[38]
Sole	*Solea solea*	[68]
Silver carp	*Hypophthalmichthys molitrix*	[63]
Sturgeon	*Acipenser baeri*	[77]
Turbot	*Scophthalmus maximus*	[68]
Zebrafish	*Danio rerio*	[78]

**Table 3 genes-10-00700-t003:** Selected non-fish species susceptible to *Y. ruckeri* infections (also includes random water and sewage sampling).

Name	Scientific Name	Reference
Muskrat	*Ondatra zibethica*	[21]
Eurasian otter	*Lutra lutra*	[60]
Sea gulls and other birds	*Larus spp.*	[79]
	*Falco spp.*	[39]
Humans (wound infection)	*Homo sapiens*	[80]
Common mudpuppy	*Necturus maculosus*	[81]
Turtles	*Cheloniidae*	[39]
Water Sewage sludge	-	[82]
Aquatic invertebrates	-	[79]

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
