# Peer review of "Overcoming Fish Defences: The Virulence Factors of Yersinia ruckeri"

_genes, 2019, doi:10.3390/genes10090700_

Round 1

Reviewer 1 Report

This review is well writing and easy to follow. This manuscript can be processed as it is. 

Author Response

thank you!

please see the replies to referee 2 and 3 for your information

Reviewer 2 Report

'Overcoming fish defences: the virulence factors of Yersinia ruckeri' by A. Wrobel, J. Leo, and D. Linke, submitted to Genes.

This is a well-written, informative overview of the current knowledge of the virulence factors of the bacterium Yersinia ruckeri, the causative agent of enteric redmouth disease (ERM) in many types of fish.  This disease is an increasing problem particularly in modern-day fish aquaculture.  Better insight into the precise mechanisms by which Y. ruckeri evades the various fish defenses is badly needed, and the current paper makes a useful contribution by reviewing the current state of knowledge.

Overall, I have no major criticisms.  As one minor issue, the color designations in the Fig. 1 and its Legend need to be reviewed carefully.  For example, the T4P system is green in the Figure, but indicated to be dark blue in the Legend.  TheT6SS system is blue in the Fig., but green in the Legend.  Please check all other designations carefully.  What is the orange 'square' in the upper outer membrane?  Also, the ylhBA directly above it looks like a squashed insect to me.  This can be improved.

Author Response

Thank you for the friendly comments. The color scheme of figure 1 was indeed skewed, and we fixed the figure _and_ the figure legend accordingly. We also replaced the "squashed bug" :-). See also the responses to referee 3 for more detail. 

Reviewer 3 Report

Dear Authors

I found the review interesting and endorse its publication after minor edits. Please check carefully all the corrections and comments suggested as "comments" on the attached PDF. I think it would be very useful if you could discuss the link between the current serotyping scheme and the correspondent virulence factors. 

good effort 

Author Response

Thank you for the insightful comments and especially for the small corrections all over the manuscript that were suggested

We have now edited the whole manuscript for small errors according to referee 3. Specifically, some references were added in the right places for clarification, and some minor text edits were done to clarify some of the questions that were asked by referee 3. We have also edited/corrected the figure and color scheme of the figure (also noted by referee 2). We have added a new table that includes also non-fish species affected by Y. ruckeri. 

Two suggestions of referee 3 were not followed:

we did not find it useful to put all the standard metabolic/microbiology tests in a table as they are not highly relevant to the rest of the text (but we did edit the text there a bit for clarity) we have not added original photos of diseased fish as we do not have access to any (we do not work with live fish ourselves, and did not want to reproduce pictures from authors of other publications here). But we do refer to literature with such pictures in multiple places.